# Simulating Performance of CHIMERE on a Late Autumnal Dust Storm over Northern China

**Siqi Ma** [1,2], **Xuelei Zhang** [1,*], **Chao Gao** [1,2], **Quansong Tong** [1,*], **Aijun Xiu** [1], **Hongmei Zhao** [1] and **Shichun Zhang** [1]

1    Key Laboratory of Wetland Ecology and Environment, Northeast Institute of Geography and Agroecology, Chinese Academy of Sciences, Changchun 130102, China; masiqi15@mails.ucas.ac.cn (S.M.); gaochao@iga.ac.cn (C.G.); xiuaijun@neigae.ac.cn (A.X.); zhaohongmei@neigae.ac.cn (H.Z.); zhangshichun@neigae.ac.cn (S.Z.)
2    University of Chinese Academy of Sciences, Beijing 100049, China
*    Correspondence: zhangxuelei@neigae.ac.cn (X.Z.); tongquansong@neigae.ac.cn (Q.T.)

**Abstract:** The accurate forecasting of dust emission and transport is a societal demand worldwide as dust pollution is part of many health, economic, and environment issues, which significantly impact sustainable development. The dust forecasting ability of present air quality forecast systems is mainly focused on spring dust events in East Asia, but further improvement may be needed as there is still difficulty in forecasting autumn dust activities, such as failing to predict the serious dust storm that occurred on 25 to 26 November 2018. In this study, a state-of-the-art air quality model, CHIMERE, with three coupled dust schemes was introduced for the first time to simulate the dust emissions during this event to qualitatively and quantitatively validate its dust simulating performance over Northern China. The model results reported that two of the three dust schemes were able to capture the dust emission source located in Gansu Province and reproduce the easterly dust transport path, showing moderately close agreement in the horizontal and vertical distribution patterns with the ground-based and satellite observations. The simulated $PM_{10}$ concentration had a better relationship with the observed values with a correlation coefficient up to 0.96, while it was lower in the transported areas. Meanwhile, the simulations also presented incorrect dust emission positions such as in areas between the Hulun Buir sandy land and Horqin sandy land. Our results indicate that CHIMERE exhibits reasonably good performance regarding its dust simulation and forecast ability over this area, and its application would help to improve the dust analysis and forecast abilities in Northern China.

**Keywords:** autumnal dust storm; Northern China; numerical modeling; CHIMERE; performance evaluation

## 1. Introduction

Wind-blown dust is one of the primary components of total atmospheric aerosol loading [1]. It is usually emitted from areas with high wind speed and dry and erodible surfaces, such as the desert, barren locations, and cropland areas. During its lifetime in the atmosphere, wind-blown dust has a significant impact on the Earth-Atmospheric system [2], air quality [3], atmospheric visibility [4], ecosystem nutrient deposition [5], and even poses great threats to human health [6]. The Sahara desert in North Africa [7], alluvial plain and deserts in West Asia and Central Asia [8], deserts and sandy lands in East Asia [9], various erodible landforms in Australia [10], and deserts in southwest USA [11] etc. are hot-spot dust areas. The estimated emission of mineral dust with diameters below 10 μm is reported to be between 1000 and 4000 Tg yr$^{-1}$ on a global scale [11]; meanwhile, there are about 800 Tg yr$^{-1}$



of Asian dust emissions injected into the atmosphere annually [12]. Therefore, it is very important to understand the occurrence and intensity of dust events, as well as their spatial–temporal distribution.

Numerical modeling is an effective way to not only help understand and develop the mechanism of wind-blown dust emission, but also to predict its activities and effects on air quality and climate. Research on wind-blown dust and numerical dust forecasting has received a much increased focus during the last few decades [13,14]. Many global and regional models and forecast systems have coupled various dust emission schemes and provided dust analyses and forecasts over the past 20 years, such as GOCART [15], SC-DREAM8b [16], CARMA-dust [17], WRF-Chem [18], CMAQ [19], CHIMERE [20], LOTOS-EUROS [21], NAQFC [22], NAQPMS [23], CFORS [24], and CUACE/Haze [25]. These models are conducted to forecast dust activities, for comparison and validation with observations for dust events and long-term aerosol variabilities, and are also widely used to study the effects of transported dust particles on air quality and climate, all of which are dedicated to serve sustainable human development.

Springtime is considered to be have a high incidence of dust events in East Asia due to the low soil moisture, strong wind velocity, and relatively sparse vegetation [26]. The modeling analyses and evaluations usually focus on dust activities during this period [27,28]. However, dust storms also tend to happen in the cold seasons, such as autumn and early winter which are thought to be the period of least occurrence [29,30], and the research on cold-season dust storms has been neglected. During 25 to 26 November 2018, a severe dust storm occurred in Northwestern China which was transported over a long distance, extending to North China, seriously impacting human activities in these affected areas [31]. Nevertheless, no open air quality numerical forecasts were able to predict this dust event, which demonstrates the insufficient ability of the present air quality forecast systems for the prediction of dust emission and transport.

The objective of this study is to firstly validate the modeling performance during the dust event which occurred in late autumn 25 to 26 November 2018 over Northern China using the CHIMERE model, a state-of-the-art air quality model which has been validated to have relative better dust-prediction performances in Europe [32], North African [33], and the Arabian Peninsula [34]. For this purpose, all the three dust emission schemes in this model are applied for the simulation and then qualitative and quantitative validations of the simulated $PM_{10}$ are performed using satellite-based products (MODIS DB and CALIPSO vertical profiles) and ground-based air quality observations.

## 2. Method

### 2.1. Study Area and Model Domain

Northern China is located at the eastern part of the global dust belt [35] and includes various deserts and desertified areas [36,37], such as the Taklimakan desert in the west, Gobi desert in the mid-west between China and Mongolia, Horqin sandy land, and Hulun Buir sandy land in the east (Figure 1). It has been defined as having a moderate and subtropical monsoon climate with prevailing westerly and northwesterly winds, which means that the dust emitted from these erodible areas often has a significant influence on large areas of China.

The research domain in this study was centered on 41.47° N, 102.02° E. It was defined on a lambert conformal projection with true latitudes of 30° and 60°, and composed of 167 × 104 grid points with a horizontal grid resolution of 36 km × 36 km and 30 vertical levels. The domain covered Northern China, as well as Mongolia, and the whole Tibetan Plateau was also included in view of the effect of its special topography on numerical simulation (Figure 1).

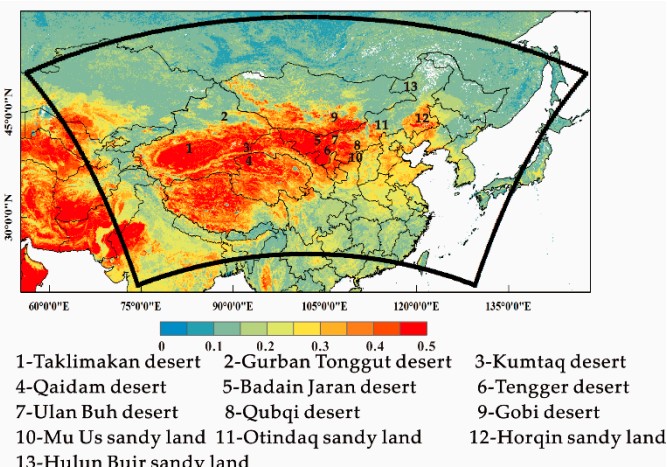

1-Taklimakan desert    2-Gurban Tonggut desert    3-Kumtaq desert
4-Qaidam desert        5-Badain Jaran desert       6-Tengger desert
7-Ulan Buh desert      8-Qubqi desert              9-Gobi desert
10-Mu Us sandy land    11-Otindaq sandy land       12-Horqin sandy land
13-Hulun Buir sandy land

**Figure 1.** The MODIS erodibility distribution, locations of primary deserts and sandy lands, and the model domain (black frame).

### 2.2. Configuration of CHIMERE

CHIMERE is an open-access multi-scale Eulerian off-line chemistry-transport model (CTM) which is mainly used to produce hourly concentrations of aerosol and pollutant gas species with horizontal resolution ranging from local to continental scales [32,38]. The aerosol module has been implemented into CHIMERE since 2004, with further modifications concerning the natural dust emissions and resuspension over Europe [39]. The evaluations of the simulation performance for long-distance transported dust have already been implemented and compared with field measurements [40,41]. In the present model version, there were three available dust emission schemes, which were established depending on Marticorena and Bergametti (1995) (MB dust scheme) [42], Alfaro and Gomes (2001) (AG dust scheme) [43] and Kok et al. (2014) (KOK dust scheme) [44], respectively. A dust production model (DPM) which extends to the global scale is available since the model version of chimere2016a and the KOK scheme, which has not been used in other air quality models before, is newly introduced at the same time.

In MB and AG dust scheme, the vertically integrated saltation flux was estimated by using the horizontal dust flux equation obtained from White [45]:

$$F_h(D_p) = K \frac{\rho_{air}}{g} u_*^3 (1 - \frac{u_{*t}}{u_*})(1 + \frac{u_{*t}}{u_*})^2 \tag{1}$$

where $K$ is a constant which equals 1 and the air density ($\rho_{air}$) is considered as 1.227 kg m$^{-3}$ following the parameterizations of Marticorena and Bergametti [42]. $u_*$ is the friction velocity calculated using the roughness length, $z_0$, and $u_{*t}$ is the threshold friction velocity, depending on the soil particle diameter size $D_p$ and $z_0$. The flux is calculated only if $u_* > u_{*t}$. Then the corresponding vertical flux in the MB dust scheme is estimated by using vertical-to-horizontal dust flux ratio ($\alpha$) with a constant value of $2 \times 10^{-6}$ and is then projected into three modes (fine, coarse, and big modes) using constant percentages (0.2, 0.6, and 0.2). As for the AG scheme, instead of being defined as a constant, $\alpha$ is calculated based on the partitioning of the kinetic energy of individual saltating aggregates and the cohesion energy of the populations of dust particles, which is written as [43]

$$\alpha(D_p) = \left(\frac{\pi}{6}\right) \rho_p \beta \sum_{i=1}^{3} \frac{p_i(D_p) d_i^3}{e_i} \tag{2}$$

This scheme assumes that dust emitted by sandblasting is characterized by three modes whose proportion depends on the friction velocity. In Equation (2), $e_i$ and $p_i$ are the cohesion energy and fractions of the kinetic energy for each mode. $\rho_p$ is the particle density (2.65 g cm$^{-3}$), $d_i$ is the mean

mass diameter and $\beta$ = 16,300 cm s$^{-2}$. In addition, a numerically optimized version of the AG dust scheme improved by Menut et al. [46] is also available in the model.

Unlike the MB and AG dust schemes, the vertical dust flux in KOK was acquired directly without converting from horizontal flux to vertical flux [44]

$$F_d = C_d f_{bare} f_{clay} \frac{\rho_a \left(u_*^2 - u_{*t}^2\right)}{u_{*st}} \left(\frac{u_*}{u_{*t}}\right)^{C_\alpha \frac{u_{*st} - u_{*st0}}{u_{*st0}}}, \tag{3}$$

This scheme is controlled by some dust emission coefficients, such as the bare soil fraction ($f_{bare}$), soil clay fraction ($f_{clay}$), and dust emission coefficient $C_d$, that represents the soil erodibility, and $C_\alpha$, which is a dimensionless coefficient defined as 2.7. Besides $u_{*t}$, two parameters relevant to it are also introduced. $u_{*st}$ is the standardized threshold friction velocity as a value of $u_{*t}$ at a standard atmospheric density (1.225 kg m$^{-3}$) and $u_{*st0}$ represents $u_{*st}$ for an optimally erodible soil which was chosen as 0.16 m s$^{-1}$ [44]. In addition, there is only one dust emission mode in this scheme rather than three in the MB and AG schemes. The methods for calculating the parameters used for each scheme in the CHIMERE model are summarized in Table 1 and detailed algorithms are provided in Part 1 of the Supplementary Materials.

**Table 1.** Information of algorithms for dust emission calculation.

|  | Dust Emission Flux | Vertical-To-Horizontal Dust Flux Ratio | Threshold Friction Velocity | Friction Velocity |
|---|---|---|---|---|
| MB | White (1986) [45]; Marticorena and Bergametti (1995) [42] | $2 \times 10^{-6}$ | Iversen and White (1982) [47]; Shao and Lu (2000) [48] | Marticorena and Bergametti (1995) [42] |
| AG | White (1986) [45]; Marticorena and Bergametti (1995) [42] | Alfaro and Gomes (2001) [43] | | |
| KOK | Kok et al. (2014) [44] | — | | |

In this study, the Weather Research and Forecasting (WRF) model version 3.9.1 was used to conduct the meteorological simulations and provide the meteorological output fields to drive the CHIMERE model. The physical parameterization schemes used in the WRF model were WRF Double-Moment 6-class microphysics scheme, the Rapid Radiative Transfer Model for Global climate models (RRTMG) longwave and shortwave radiation scheme, the Pleim–Xiu scheme for surface layer and land-surface schemes, the Asymmetric Convective Model with non-local upward mixing and local downward mixing (ACM2) boundary layer scheme, and the Grell–Devenyi ensemble cumulus scheme. As the Global Forecast System (GFS) forecast products of the National Center for Environmental Prediction (NCEP) were often used as a meteorological input during the numerical forecast, data with a horizontal resolution of 0.5° × 0.5° (available every 6 h) were introduced to act as the initial and boundary fields in this study (https://www.ncdc.noaa.gov/data-access/model-data/model-datasets/global-forcast-system-gfs). Following this, the CHIMERE model, version 2017r4, would run in the MELCHIOR chemistry mechanism with all the three dust emission schemes. Iversen and White [47] and Shao and Lu [48] methods (IW and SL hereafter) were chosen as algorithms for calculating $u_{*t}$. The Emission Database for Global Atmospheric Research, Hemispheric Transport of Air Pollution (EDGAR-HTAP) global emission inventories including annual and monthly emission data for the year 2010 [49] were introduced as the anthropogenic emissions and pre-processed by the emiSURF program (version 2016b) before being incorporated into the model domain. The outermost boundaries of the meteorological driving field were removed, and there were 166 × 103 grid points in the domain of CHIMERE. The vertical levels of the model were increased from 8 (surface to 500 hPa) to 15 (surface to 200 hPa) to obtain more detailed vertical results. All the simulations were run for 96 h from 24 to 27 November 2018, and the first 24 h was regarded as the model spin-up time.

Additionally, unlike other air quality models, several static land-cover datasets were also used for the dust emission calculation in this model, and these datasets were simultaneously prepared during model installation. One dataset used was the USGS land-use dataset with a high horizontal resolution ($0.0083° \times 0.0083°$), and four kinds of land-use types—barren land, shrub land, grassland, and cropland—are regarded as erodible types (Figure 2a). The monthly USGS vegetation coverage (green fraction) (Figure 2b) with the same horizontal resolution as the land-use type and MODIS erodibility (Figure 1) ($0.05° \times 0.05°$) also acted as input data. These three datasets (land-use type, vegetation coverage and erodibility) were used for the calculation of erodible fraction of which default algorithm in the model was described as when the land-use type was cropland and grassland, the erodible fraction depended on the fraction of the area without vegetation coverage, and it equaled the MODIS erodibility for the land-use type of shrub land and barren land (Figure 2c). Note that only the USGS dataset was available for this calculation. As to the roughness length, which takes part in the dust flux calculation, the GARLAP (Global Aeolian Roughness Lengths from ASCAT and PARASOL) dataset ($0.0083° \times 0.0083°$) derived from LiDAR and satellite observations was used (Figure 2d).

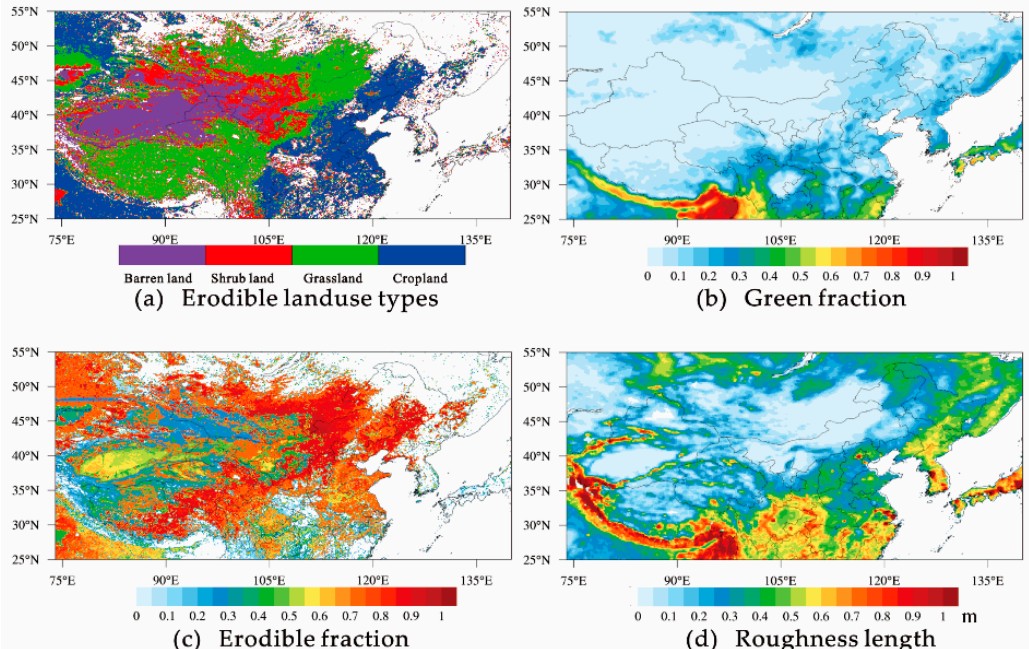

**Figure 2.** Geographic information of the datasets used in CHIMERE. (**a**) Erodible land-use types, (**b**) green fraction and (**c**) erodible fraction in November, and (**d**) roughness length distribution.

### 2.3. Observational Data Sources

The ground-based monitored air quality data were obtained from the website of the national air quality history database since 13 May 2014 (http://beijingair.sinaapp.com), which collects air quality data and information from the China national environmental monitoring center, which updates weekly. The hourly monitoring data used in this study included the concentrations of particulate matter ($PM_{10}$ and $PM_{2.5}$) in the cities with a latitude higher than $30°$ N within the time period ranging from 24 to 27 November 2018.

In order to obtain further knowledge of the dust intensity and transport path with a regional overview of this dust event, the Deep Blue Aerosol Optical Depth (AOD) with a resolution of $1° \times 1°$ from MODIS-Aqua production was obtained from the NASA Giovanni website at https://giovanni.gsfc.nasa.gov/giovanni. Furthermore, two data products of the Cloud-Aerosol LiDAR and Infrared Pathfinder Satellite Observation (CALIPSO) were also used to retrieve the vertical distributions and types of the aerosols, that is, level 2 vertical feature mask (VFM) data and level 2 aerosol profile data with a resolution of 5 km (version 3.4, https://calipso.larc.nasa.gov).

## 3. Results

### 3.1. Dust Event during 25 to 26 November 2018

A severe dust storm occurred and influenced a large area of Northern China during 25 to 26 November 2018. According to the satellite and ground-based observations, the dust was emitted from the mid-northern part of Gansu Province with a daily mean $PM_{10}$ concentration exceeding 1500 µg m$^{-3}$, while its eastern areas still maintained a relative low $PM_{10}$ (<300 µg m$^{-3}$) and most of the regions, especially the Jing-Jin-Ji region, were mainly affected by anthropogenically sourced fine particle ($PM_{2.5}$) emissions (Figure 3a,d,g). At the same time, the Gurban Tonggut desert in Xinjiang Province also presented a higher dust concentration. On the 26 November, the $PM_{10}$ concentration increased and constituted a higher-intensity belt in the eastern areas of the dust source position, along Gansu, Shaanxi, Inner Mongolia, Shanxi Province, extending eastward and southward to Liaoning and Henan Province, respectively. The $PM_{10}$ concentrations in this belt area almost saw a two-fold increase (Figure 3e). This demonstrated that the produced dust moved east, and the dust storm, which happened in Northern China, might be caused by long-distance and trans-regional dust transport. Furthermore, a higher AOD (0.7–1.1) appeared in the central area between the Inner Mongolia Province and south Mongolia, which also indicated that there might be another dust source in the Gobi Desert (Figure 3b). Finally, on 27 November the dust intensity reduced back to the situation before the event (Figure 3f). Meanwhile, the vertical distributions of the aerosol subtypes derived from CALIPSO observation indicated that dust filled the atmosphere below 3 km in the area between 40–50° N and acted as the primary pollutant during the dust event (Figure 3j,k).

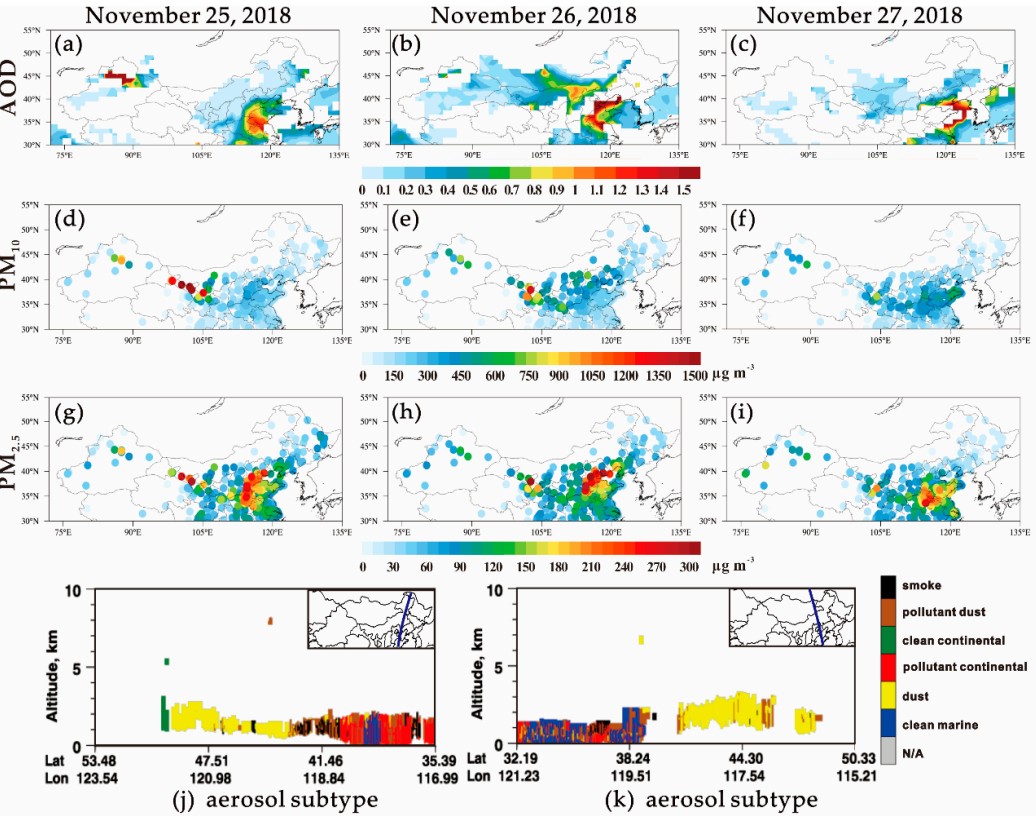

**Figure 3.** Daily distributions of MODIS aerosol optical depth (AOD) at 550 nm (**a**–**c**), and the $PM_{10}$ (**d**–**f**) and $PM_{2.5}$ (**g**–**i**) concentrations at the near surface on 25 to 27 November 2018. Vertical CALIPSO profiles of classified aerosol subtypes at 18:00 Coordinated Universal Time (UTC) on 25 November (**j**) and at 5:00 UTC on 26 November (**k**). The colored dots in (**d**–**i**) indicate the PM concentrations observed from ground-based air quality monitoring stations.

In addition, the concentrations of the observed $PM_{10}$ and $PM_{2.5}$ during this event also showed that, during this event, the mass of larger particles with a radius between 2.5 and 10 μm were significantly higher than that of fine particles. The ratios between the $PM_{2.5}$ and $PM_{10}$ in the dust source areas were generally below 0.2 and the mean ratios decreased from 0.60 to 0.41 in northern regions during the dust event, indicating that the particulate matters (PMs) were dominated by large dust particles which coincided with the CALIPSO observations.

### 3.2. Horizontal Distribution of Simulated Dust

For the sake of analyzing the model performance on different stages of the dust activities, Figure 4 provides the distributions of the observed and simulated $PM_{10}$ concentration in Northern China during the periods of the start of emissions (8:00 UTC 25 November), transporting (1:00 UTC 26 November), and the longest transport ranges to east (8:00 UTC 26 November) and south (18:00 UTC 26 November). This showed that the results simulated by using MB and AG dust emission scheme with the $u_{*t}$ method of IW were capable reproducing the dust source position when the dust emitted from the central area of Gansu province and the $PM_{10}$ in the Jing-Jin-Ji region also had higher concentrations, which were in accordance with the observations (Figure 4a–c). However, they did not show the dust emissions in the north of Xinjiang Province. During the transport stage, these two dust schemes could also illustrate the eastward dust path and the higher $PM_{10}$ concentration in the mid-east of Inner Mongolia, which might be attributed to the dust emission from the Otindaq sandy land. However, no dust emission between the Horqin and Hulun Buir sandy land could be observed, while it appeared in the simulated results (Figure 4e–h). In the later period of this event, when dust was transported for long distances to the east, the discrepancies between the observed and simulated $PM_{10}$ distribution were significant. The simulations were able to reproduce the distribution patterns of the south-end of the transport path; however, the concentrations were largely underestimated. Furthermore, unlike the observations, which showed that dust intensity remained high in the western areas, the simulated $PM_{10}$ remarkably decreased almost to its background values. Meanwhile, considerable dust emission occurred in the northeast of China, which was inconsistent with the actual situation (Figure 4j,k,n,o). In comparison, during the dust event, the $PM_{10}$ distribution simulated by the KOK scheme provided no discernible information about this dust event in the period of starting to emit dust and the high-concentration area located in Eastern China, yet this was more southernly than the observation. In fact, the variation and distribution of modeling $PM_{10}$ using the KOK scheme was quite similar to those simulated by anthropogenic emission alone, which meant that results calculated via this scheme had not presented reasonable dust emission and transport patterns during this event (Figure 4d,h,l,p). For the simulated results using $u_{*t}$ of SL, the areas of the dust source and dust concentration were lower than those with IW, while the areas of dust distribution were relatively similar, which indicated that dust was more difficult to emit with this method. However, there was no further difference between them.

On the other hand, another striking distinction of the observations and modeling results was their concentration levels. The $PM_{10}$ calculated by MB and AG schemes had similar spatial–temporal distribution patterns, but their values showed large differences. Simulated concentrations of AG were comparable with the observations, while those of MB were considerably overestimated, at about four times higher than the observed values. In addition, it is important to note that the simulated dust concentration reduced more rapidly than the observations in the transported areas, and this resulted in understimations in regions such as the north of Central China.

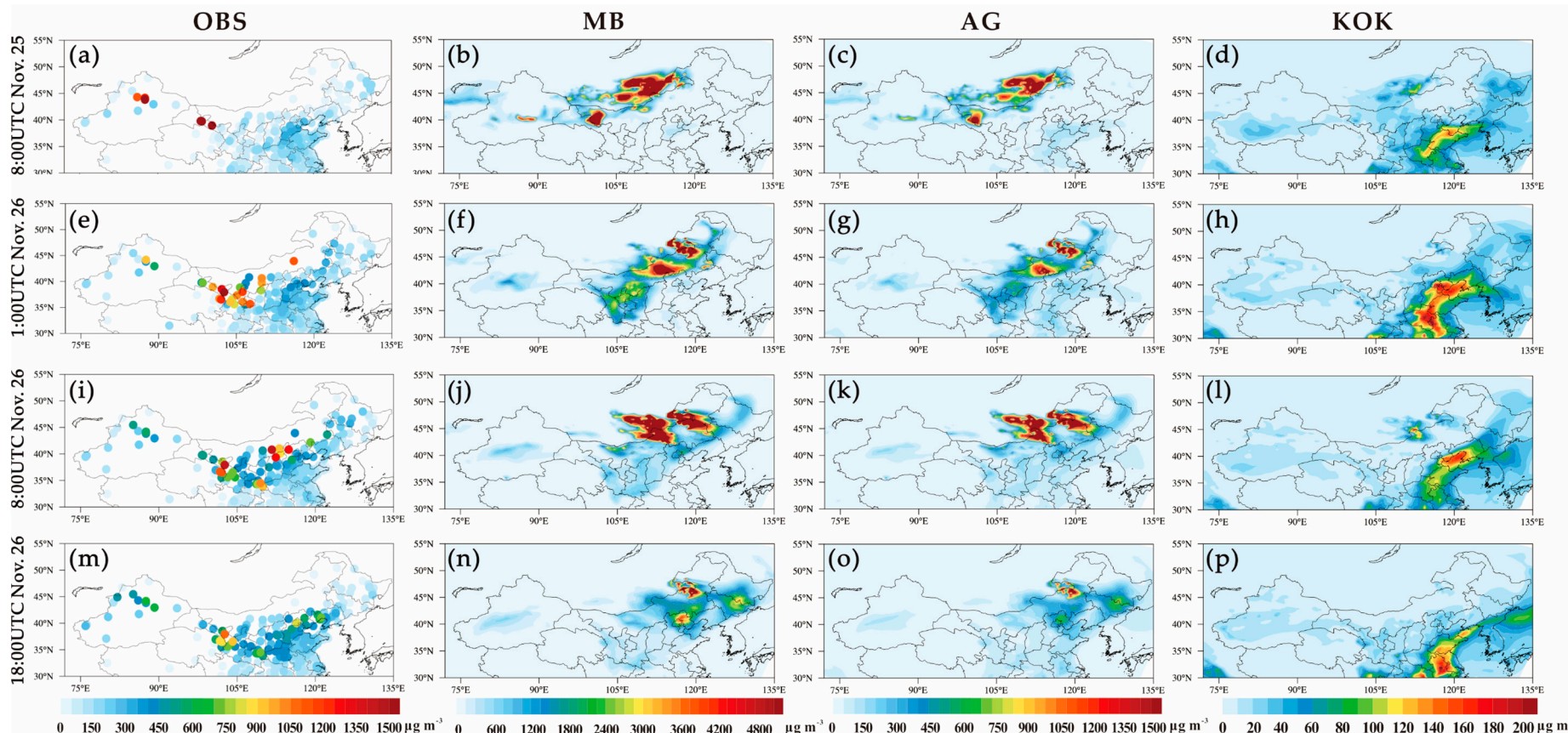

**Figure 4.** Distributions of site-observed and simulated PM$_{10}$ concentration using MB, AG, and KOK dust schemes with Iverson and White (IW) at 8:00 UTC 25 November (**a–d**), 1:00 UTC 26 November (**e–h**), 8:00 UTC 26 November (**i–l**), and 18:00 UTC 26 November (**m–p**).

### 3.3. Vertical Distribution of Simulated Dust

As the model output did not include the variable of extinction coefficient in CHIMERE, the vertical distributions of the $PM_{10}$ mass concentrations obtained by using MB and AG scheme were compared with the CALIPSO aerosol extinction coefficient at 18:00 UTC on 25 November (left panel) and at 5:00 UTC on 26 November (Figure 5). As displayed in these figures, the extending height of the dust plume and the vertical variation with altitude were able to be described in the simulations. The dust plume, which reached the height of about 2 km along the satellite moving path from 43.00° N, 119.35° E to 30.70° N, 115.71° E on 25 November, could be both observed in the CALIPSO image and simulation of the two schemes (Figure 5a,c,e). However, the modeling concentration under 1 km between 30° N and 40° N was not as significant as the observation. In comparison, the dust simulated by MB in higher latitude (45–50° N) presented a slightly larger intensity and reached the height of 4 km, while the result of AG was much closer to the observation. The similar large dust concentration with higher vertical extension in high latitude was fairly clear in the simulations of 26 November as well (Figure 5b,d,f). Their vertical distributions also indicated dust overestimation and possibly incorrectly simulated dust source positions in the northeast of China, which was also found in the horizontal comparison. Moreover, both of the schemes exhibited weak reproduction for the areas south of 35° N. As for the KOK scheme, its results were still less intense and showed no comparable vertical patterns with the CALIPSO observations (Figure S1).

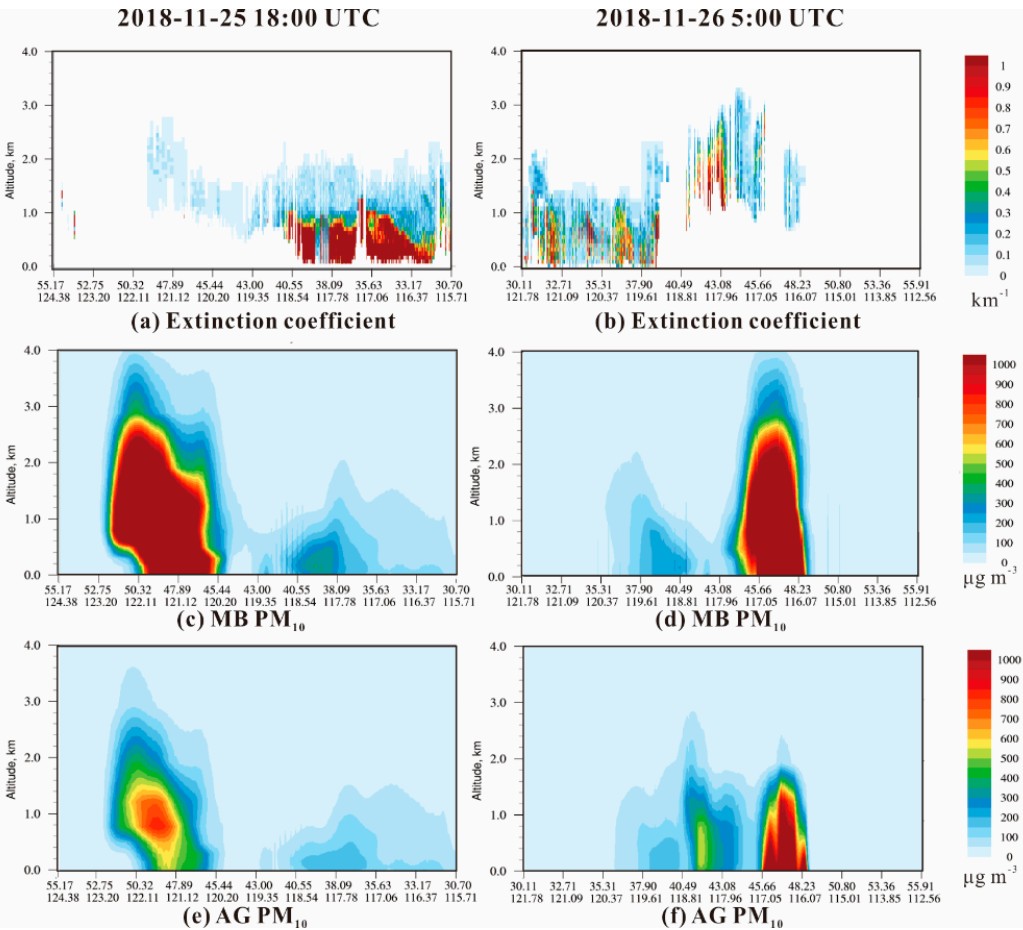

**Figure 5.** Vertical distributions of aerosol extinction coefficient from the CALIPSO and modeling $PM_{10}$ concentrations simulated by MB and AG dust scheme at 18:00 UTC on 25 November (**a,c,e**) and at 5:00 UTC on 26 November (**b,d,f**).

### 3.4. Quantified Validation

In order to further validate the model performance with regard to the ability to reproduce dust emissions, four statistical parameters—correlation coefficient (CORRE), relative mean square error (RMSE), normalized mean bias (NMB), and normalized mean error (NME)—of simulations and observation data in 246 ground-based air quality sites in Northern China were collected for quantitative comparisons. The values of four parameters varying with latitude and their statistical results are provided in Figure 5 and Table 2. We found that the CORREs between the observed and simulated $PM_{10}$ widely fluctuated with latitude (Figure 6a). The values could be divided into three parts: A relative lower correlation between 30–32° N, an increase between 32–42° N, and a decrease when the latitude was higher than 42° N for MB and AG schemes. However, the CORRE variation for KOK was nearly the opposite. It indicated that the model with the MB and AG dust schemes had better performance in the dust emission areas with CORRE up to 0.96, while it decreased in transported areas with average values of 0.27 (MB) and 0.17 (AG). The regional mean CORREs of these two schemes were 0.30 and 0.21, passing the significant test of 95% and 90%, respectively. However, the result of the KOK scheme demonstrated terrible dust simulating ability in this area, with a negative CORRE of −0.17.

**Table 2.** CORRE, RMSE, NMB, and NME between the simulations of MB, AG, and KOK scheme and measurements $PM_{10}$ in the source area, transported area, and the whole area of Northern China.

|  |  | **MB** | **AG** | **KOK** |
|---|---|---|---|---|
| | Source area | 0.76 | 0.74 | −0.13 |
| CORR | Transported area | 0.27 | 0.17 | −0.15 |
| | Regional mean | 0.30 | 0.21 | −0.15 |
| | Source area | 623.04 | 553.49 | 679.82 |
| RMSE | Transported area | 252.69 | 156.59 | 188.78 |
| | Regional mean | 276.88 | 182.51 | 220.85 |
| | Source area | 0.15 | −0.75 | −0.96 |
| NMB | Transported area | 0.85 | −0.36 | −0.69 |
| | Regional mean | 0.80 | −0.39 | −0.71 |
| | Source area | 78.51 | 77.91 | 95.59 |
| NME (%) | Transported area | 121.62 | 59.34 | 71.20 |
| | Regional mean | 118.81 | 60.56 | 72.79 |

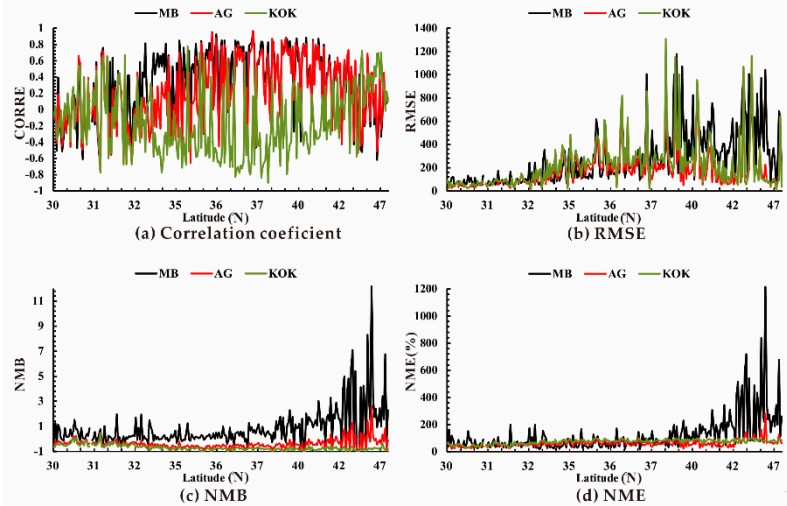

**Figure 6.** Correlation coefficient (CORRE) (**a**), relative mean square error (RMSE) (**b**), normalized mean bias (NMB) (**c**), and normalized mean error (NME) (**d**) between the simulations of MB, AG, and KOK scheme and observed $PM_{10}$.

Meanwhile, all schemes had lower bias and errors with latitude below 38° N (for RMSE) or 42° N (for NMB and NME) and dramatically rose with the increase of latitude, which also showed higher discrepancies between the observations and simulations in the northeast areas (Figure 6b–d). Among these three dust emission schemes, AG showed the lowest RMSE (mean value of 182.51), NME (60.56%) and had the NMB closest to 0 (−0.39, Table 2), meaning that it had the best reproducing ability during this dust event in Northern China.

## 4. Discussion

The results of the model comparisons against ground-based and satellite observations showed that the $PM_{10}$ concentrations modelled by CHIMERE using different dust schemes were quite different. The MB and AG schemes had similar spatial distribution patterns and were capable of capturing the dust source of this autumnal dust event. However, both of them showed incorrect dust emitting positions in the northeast of China leading to large bias and errors. The results simulated by MB produced non-negligible overestimations and AG reported slight underestimations, especially in the transported area. For the KOK scheme, the dust activity was unreasonably simulated during this event and presented nearly no similarity with observations.

There were several parameters controlling the dust emission, such as the horizontal dust emission flux, $\alpha$, erodible fraction, $u_{*t}$, and $u_*$. The MB and AG scheme used the same horizontal dust flux formula and the same algorithms of $u_{*t}$ and $u_*$, and so their large discrepancies could be attributed to $\alpha$. In the AG scheme, the cohesive/binding energy of the dust particles and the kinetic energy were taken into account when establishing the formula of $\alpha$. The number flux of the dust particles was proportional to the ratio between the kinetic energy loss and the binding energy, which was higher for small dust particles [50]. Therefore, $\alpha$ was sensitive to the diameter and speed of the saltating soil grains. The calculated $\alpha$ value ranged from $2 \times 10^{-8}$ to $2 \times 10^{-6}$ and mainly fluctuated between $10^{-7}$ and $10^{-6}$ [43]. This was about two times smaller than the constant $\alpha$ ($2 \times 10^{-6}$) used in the MB scheme. Likewise, the $\alpha$ calculated based on $D_p$ and $u_*$ in the AG scheme was much lower than those derived from soil types which ranged from $10^{-6}$ to $10^{-2}$ (Table 3, Figure S2). However, due to the use of the mass median diameter instead of mass mean diameter in Alfaro and Gomes [43], the dust mass flux would be overestimated by a factor of about 2.5 [51]. Nevertheless, the evaluation showed the AG scheme gave the closest dust concentration to the observation, which indicated that the higher $\alpha$ used in the MB scheme might be the reason for the critical overestimation in this area. On the other hand, although the AG dust scheme had reported the best reproduction capacity, there were still issues during the modeling calculation. According to the methodology reported in Alfaro et al. [52], the equations used to obtain kinetic energy were divided into two parts: One was used when $u_* < 0.27$ m s$^{-1}$ and the other one was for $0.27$ m s$^{-1} < u_* < 0.55$ m s$^{-1}$ (Equation S2, S3). However, the latter one was not introduced into the model, which meant the kinetic energy might be incorrectly calculated when $u_* > 0.27$ m s$^{-1}$.

Regarding the unsatisfactory performance of the KOK scheme, there are three possible reasons for this. The first one was that, unlike the sandblasting theory used in the other two schemes, the KOK scheme was built on the foundation of fragmentation theory, which was depicted as dust aggregates fragmented by saltators into smaller particles and vertically emitted to the atmosphere [44]. This might be not applicable in a region with large areas of grassland, shrub land, and cropland enriching organic matter [53], which are different from desert and barren land. The second might be ascribed to the dust emission coefficient $C_d$ that scaled the dust emission flux in a way which was similar (but not identical) to the dust source functions. However, in the region of East Asia, it only covered parts of Mongolia and Inner Mongolia of China on the basis of the global normalized $C_d$ distribution [54]. This could explain the failure to reproduce the dust emitting position, which was located outside the coverage of $C_d$, in this event. Furthermore, its value was lower compared to the normalized Ginoux source function and this could result in the underestimation of the dust flux calculated by the KOK scheme. Finally, according to the source code in the CHIMERE model, the dust flux was multiplied by the

erodible fraction after being scaled by $C_d$, and this meant the erodible factor might be multiplied twice during the calculation, making the result much smaller than the observations.

**Table 3.** The variation ranges of the vertical-to-horizontal dust flux ratios ($\alpha$) of MB and AG schemes, and another four frequently-used algorithms.

|  | Parameters | Variation Range (cm$^{-1}$) |
|---|---|---|
| MB scheme |  | $2 \times 10^{-6}$ |
| AG scheme | $D_p, u_*$ | $2 \times 10^{-8}$–$2 \times 10^{-6}$ |
| Marticorena and Bergametti [42] | clay fraction | $10^{-6}$–$2 \times 10^{-4}$ |
| Lu and Shao [55] | $u_*$, soil types | $10^{-5}$–$2 \times 10^{-2}$ |
| Shao [56] | $u_*$, soil types | $10^{-5}$–$2 \times 10^{-2}$ |
| Foroutan et al. [57] | $u_*$, soil types | $10^{-5}$–$2 \times 10^{-4}$ |

Additionally, the dust-emitting positions from areas in the east of Inner Mongolia between Hulun Buir and Horqin were not observed at nearby monitoring sites, and this led to large discrepancies with measurements. In CHIMERE, this erodible area seemed to be susceptible to dust emission, and this might be ascribed to $u_{*t}$ not performing well over grassland and shrub land, the properties of which were different from other erodible land-use types. Meanwhile, the fallen leaves in the cold season could increase the $u_{*t}$ which caused the reduction of dust emission [58], yet this effect was not considered in CHIMERE. Lastly, note that the use of $z_0$ from the GARLAP dataset in CHIMERE model would lead to lower dust flux compared to that obtained from land-use data [59].

## 5. Summary and Conclusions

In this study, a dust event simulation of a three dimensional chemistry transport model CHIMERE was conducted over Northern China for the first time. All the three dust emission schemes coupled in this model were performed for a severe dust event during 25 to 26 November 2018 and used to compare and validate their dust simulation ability in this area.

According to the comparisons and validations, we found that the PM$_{10}$ simulated by dust schemes of MB and AG in CHIMERE showed similar spatial patterns and were able to capture the primary dust source position and reconstruct the spatial distribution of this dust event. They both had better reproducing abilities in areas near to the dust source while they were weak in the dust-transported areas, and the MB scheme showed a relatively higher correlation than AG. However, the MB scheme gave considerable overestimations, and both of them had incorrect dust emission sources, such as areas between Hulun Buir and Horqin. In general, the statistic parameters demonstrated that the AG dust emission scheme had the best performance among the simulations, with significant regional mean correlations, low bias and error, and moderately close agreement with the vertical structure of CALIPSO images.

This was the first time that the CHIMERE model exhibited its dust simulation ability in Northern China, and it presented a fairly good reproduction result. Further tests of this model for more significant dust events and long-term dust variation simulations over Northern China will be conducted, and the application of CHIMERE in this area would help to improve dust analysis and forecast abilities. In addition, field measurements and experiments should also be implemented for analyzing $u_{*t}$ over land-use types, such as grassland.

**Supplementary Materials:** The following are available online at http://www.mdpi.com/2071-1050/11/4/1074/s1, Figure S1: Vertical distributions of aerosol extinction coefficient from the CALIPSO and CHIMERE simulated PM10 concentrations at 18:00 UTC on Nov. 25 (left panel) and at 5:00 UTC on Nov. 26 (right panel), Figure S2: vertical-to-horizontal dust flux ratio ($\alpha$) for sand, loam and clay as a function of friction velocity ($u_*$) following Marticorena and Bergametti (MB95), Lu and Shao (LS), Shao (S04) and Foroutan et al. (F17).

**Author Contributions:** Conceptualization, X.Z.; Data curation, S.M.; Formal analysis, S.M.; Funding acquisition, X.Z.; Methodology, S.M., X.Z. and C.G.; Software, C.G. and A.X.; Supervision, X.Z.; Validation, Q.T. and S.Z.; Visualization, S.M.; Writing—original draft, S.M.; Writing—review and editing, X.Z., Q.T., A.X. and H.Z.

**Funding:** This work was financially supported by the National Natural Science Foundation of China (NSFC) (No. 41571063 and 41771071), National key R&D Plan of China (No. 2017YFC0212304), Hundred-Talent Program (Chinese Academy of Sciences, No. Y8H1021001) and Flexible Talents Program of Jilin (No. Y7D7011001).

**Acknowledgments:** We thank the website of national air quality history database for providing the monitoring atmospheric pollutant data. We also thank Earth Observing System Data and Information System for providing CALIPSO products and NASA's Giovanni-Interactive Visualization and Analysis team for providing MODIS deep blue AOD data.

**Conflicts of Interest:** The authors declare no conflict of interest.

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
