# Peer review of "Simulating Performance of CHIMERE on a Late Autumnal Dust Storm over Northern China"

_sustainability, doi:10.3390/su11041074_

Round 1
Reviewer 1 Report
This topic is important and useful for air quality research in China. However, this draft is far from enough for a research paper, and it’s more like a very simple technical note. The originality and novelty are very low.
To publish as a research paper, the authors must:
(1) Add more detailed difference about the three schemes;
(2) Add more analysis about how the difference of the schemes affect the dust simulation.
Author Response
Response to Reviewer 1 Comments
Point 1: Add more detailed difference about the three schemes.
Response 1: Thanks for your very kind advice. According to your suggestion, we further introduced each dust scheme (MB, AG and KOK scheme) used in CHIMERE in detail and presented main dust emission formulae in Section 2.2 of the revised manuscript. Meanwhile, we also explained the primary differences of the dust schemes in this part, such as the theory for establishing the dust schemes, used parameters, as well as their algorithms and dust emission modes. In addition, the detailed formulae and methods for obtaining parameters such as friction velocity and threshold friction velocity were also provided in the supplementary information. The detailed descriptions are presented in the revised manuscript (line 92~119).
Point 2: Add more analysis about how the difference of the schemes affect the dust simulation.
Response 2: We are extremely grateful to Reviewer 1 for pointing out this problem, and very sorry for our negligence of detailed explanation for the differences of the results calculated by different schemes. On the basic of describing and comparing the simulated results in the Result section (3.2~3.4), the differences of the performances simulated by three dust schemes and their reasons were further analysed in the Discussion section. As to the results simulated by MB and AG scheme, we focused on analyzing the effects of different vertical-to-horizontal dust flux ratio (α) on the simulated dust intensity and compared them with the α calculated by other frequently-used algorithms. In addition, an issue of misuse of the equations when calculating α with AG scheme is also found out. Meanwhile, we analysed the unreasonable results simulated by KOK scheme in three aspects, namely the fragmentation theory which was used for establishing this scheme, the coverage and value of the dust emission coefficient Cd and the source code of this scheme in CHIMERE. The detailed descriptions were presented between line 290 and 322 in the revised manuscript.
Thank the reviewer again for the constructive criticisms that have helped us to improve our manuscript. We have tried our best to improve the manuscript and made changes in the manuscript. These changes will not influence the content and framework of the paper. And here we do not list the changes but marked in the revised manuscript.

Reviewer 2 Report
Review of the Manuscript ID: sustainability-430069
The paper is an interesting intercomparison between different dust schemes in CHIMERE. Numerical simulations were performed using the model chain given by the meteorological model WRF and the transport and diffusion chemical model CHIMERE. The dust event of November 25-26, 2018 was considered for comparison.
The study is carried out rigorously and the approach is correct. The authors show both qualitative and quantitative comparisons and discuss the results properly.
For these reasons I would be favourable to the publication. However, I have two main concerns. First, I strongly suggest to revise the English language, possibly from a mother tong speaker, to improve the quality and the understandability of the paper, which in many parts is, in my opinion, rather obscure. Second, since the results show the difficulty of the model to reproduce the observations, it may be useful to evaluate the model performance in this case with the one in similar studies from literature.
Some minor point
Lines 54-56 Please provide a reference
Line 60: which event in the late autumn?
Line 72: It was defined
Lines 95-98 are not clear. Please, rewrite the sentence
Line 108: CHIMERE
Line 114: what is the line circle?
Line 115: why the number of levels was changed respect to the default?
Figure 3 should be described in more detail. Why the optical depth is compared with PM? Does PM indicate the integral concentration? The Figure itself is not clear since it is too small.
Line 165: “It was showed” Please clarify how and where
Figure 3: the caption is not completed. What do indicate the dots? Are they measurement stations?
Line 175: presents
Lines 176, 177: too many parentheses
Lines 204-205: I do not understand. Please, clarify.
Figure 4 is too small. What do indicate the dots?
Line 210: The plots do not show vertical profiles (which are lines) but vertical sections
Line 211: why only two schemes are considered here?
Line 212: I wonder if it could be correct comparing extinction coefficient with PM concentration, Please, justify this choice.
Line 214 “more or less” is too vague, please be more precise.
Line 216 “weaker” respect to what? Again, I wonder if extinction coefficient and PM concentration can be compared quantitatively.
Figure 5: units are missing
Line 234: Why is it interesting to show the variation with the latitude and not with the longitude?
Line 239 (and following): what does it means “mean correlation”?
Line 269: which two schemes?
Line 304: what does it means areal mean concentration?
Author Response
Response to Reviewer 2 Comments
Point 1: I strongly suggest to revise the English language, possibly from a mother tong speaker, to improve the quality and the understandability of the paper, which in many parts is, in my opinion, rather obscure.
Response 1: Thanks for the review’s kind advice and we apologize for the English writing problem. We have examined and revised the languages carefully. And the manuscript has be already edited professionally via English Editing Service.
Point 2: Since the results show the difficulty of the model to reproduce the observations, it may be useful to evaluate the model performance in this case with the one in similar studies from literature.
Response 2: We want to thank the reviewer for the constructive and insightful advice. In the Introduction section, we briefly summarized the dust reproduction ability of CHIMERE in the areas like Europe, North African and Arabian Peninsula which presented relative better performances. However, as far as we know, most of the previous studies of model evaluation mainly focus on climate effects of different dust schemes both in global and regional scales. Comparisons and evaluations on model performances during heavy dust periods still need to be implemented. And we have already conducted a comprehensive intercomparison study in evaluating performances of different dust emission schemes in mainstream air quality models including CHIMERE and have written a paper. In this study, we mainly pay attention to the model simulation during a rare dust storm which occurred in autumn and evaluate its performance during the dust period that people seldom focus on.
Some minor point
Lines 54-56 Please provide a reference
As this dust event occurred only a few months ago, there has no published paper introducing it yet. So we provide a news from China News Service reporting it. Present lines: 54-56.
Line 60: which event in the late autumn?
The dust event occurred in late autumn during 25–26 November (late autumn) 2018, and it has been explained in revised manuscript. Present lines: 59~60.
Line 72: It was defined
Revised into “It was defined on a lambert conformal projection…”. Present line: 73.
Lines 95-98 are not clear. Please, rewrite the sentence
This part (Section 2.2) has been revised a lot and provides the information of the dust schemes more detailedly, and showed between line 91 and 118.
Line 108: CHIMERE
It has been revised into “CHIMERE”. Present line: 128.
Line 114: what is the line circle?
It means the outermost boundaries of the domain and the sentence has been revised into “The outermost boundaries of meteorological driving field was removed and there were 166 × 103 grid points in the domain of CHIMERE.” Present lines: 133~134.
Line 115: why the number of levels was changed respect to the default?
The default vertical level is from surface to 500 hPa with 8 layers, it only reaches the middle troposphere. In this study, we prepare to compare and analyse the vertical distributions of the observed and simulated results, therefore, in order to have better description of the vertical information, it needs higher and more detailed modelling vertical layers. With considering the running time of the model, the number of vertical layer is changed to 15, from surface to 200hPa (which can be regarded as tropopause). Present lines: 136~137.
Figure 3 should be described in more detail. Why the optical depth is compared with PM? Does PM indicate the integral concentration? The Figure itself is not clear since it is too small.
The caption of Figure 3 is revised into “Daily distributions of MODIS aerosol optical depth (AOD) at 550 nm (a~c), and the PM10 (d~f) and PM2.5 (g~i) concentrations at the near surface on 25–27 November 2018. Vertical CALIPSO profiles of classified aerosol subtypes at 18:00 UTC on 25 November (j) and at 5:00 UTC on 26 November (k). The colored dots in d~i indicate the PM concentrations observed from ground-based air quality monitoring stations.” And the figure size and the font size have been increased.
In this section, the spatial-temporal distributions of aerosol and PM are introduced to understand the activity of this dust event. Remote sensed AOD is not used to compare with PM. Actually, it can help us have further knowledge of the dust distribution in study region, especially the areas without ground-based monitoring stations (such as Mongolia in this study). As to the PM, it is the surface PM10 and PM2.5 concentration observed from monitoring stations, not the integral concentration.
Line 165: “It was showed” Please clarify how and where
It indicates the observed PM10 and PM2.5 concentration over the study area, and is described as “the concentrations of observed PM10 and PM2.5 during this event also showed that during this event, the mass of larger particles with radius between 2.5 and 10 μm were significantly higher than that of fine particles.” in the revised manuscript. Present line: 188.
Figure 3: the caption is not completed. What do indicate the dots? Are they measurement stations?
The coloured dots indicates the PM concentrations measured by ground-based air quality monitoring stations. And the figure information is further described in the caption.
Line 175: presents
It has been revised into “presents”. Present line: 199.
Lines 176, 177: too many parentheses
It revised into “for the sake of analyzing the model performance on different stages of the dust activities: namely starting emitting (8:00 UTC 25 November), transporting (1:00 UTC 26 November) and the longest transport ranges to east (8:00 UTC 26 November) and south (18:00 UTC 26 November).” Present lines: 200-202.
Lines 204-205: I do not understand. Please, clarify.
In the dust transported area (which is effected by the dust storm but has no dust emitting), the simulated dust concentration decreased more rapidly than the observations, which led to the underestimation in those regions. Revised into “In addition, it is important to note that the simulated dust concentration reduced more rapidly than the observations in the transported areas, and this resulted in understimations in regions such as the north of Central China.”. Present lines: 229~231.
Figure 4 is too small. What do indicate the dots?
The dots indicates the PM concentrations measured by ground-based air quality monitoring stations and the figure size and the font size have been enlarged.
Line 210: The plots do not show vertical profiles (which are lines) but vertical sections
“vertical profile” is changed to “vertical distribution”. Present lines: 237~238.
Line 211: why only two schemes are considered here?
The results shows that the KOK dust scheme presented unreasonable dust distribution, no matter in horizontal or vertical direction. It is not necessary to show the vertical distribution of the PM10 simulated by KOK scheme in the manuscript as it has very low dust intensity and presents nearly no similarity with the observations. Instead, it is provided in the supplementary information.
Line 212: I wonder if it could be correct comparing extinction coefficient with PM concentration, Please, justify this choice.
The CHIMERE output variables don’t include the extinction coefficient, and it has no vertical profile measurement for PM concentration and lacks AOD observation data in this study area. So it is hard to obtain the observed variable which is same with the simulation output. Therefore, we follow the method of Beegum et al. (2016) and compare the pattern of PM10 vertical distribution with observed extinction coefficient. We mainly want to qualitatively analyse whether the simulated dust has similar vertical pattern with observation, rather than quantitatively compare their concentrations or intensities.
Reference
Beegum, S.N.; Gherboudj, I.; Chaouch, N.; Couvidat, F.; Menut, L.; Ghedira, H. Simulating aerosols over Arabian Peninsula with CHIMERE: Sensitivity to soil, surface parameters and anthropogenic emission inventories. Atmos. Environ. 2016, 128, 185–197.
Line 214 “more or less” is too vague, please be more precise.
It is now described as “the extending height of the dust plume and the vertical variation with altitude was able to be described in the simulations.” Present lines: 240~241.
Line 216 “weaker” respect to what? Again, I wonder if extinction coefficient and PM concentration can be compared quantitatively.
“weaker” means comparing to the intensity with altitude >1 km, the simulated dust under 1 km is not as significant as the observation. The simulated result shows weaker vertical gradient than observation. And the observed extinction coefficient is used for the quantitative comparison rather than quantitative comparison. Present lines: 243~244.
Figure 5: units are missing
The units in the figures are provided.
Line 234: Why is it interesting to show the variation with the latitude and not with the longitude?
According to the analyses, we find that the dust distribution varies along the latitudes and presents a spatial pattern of higher in the north while lower in the south. Meanwhile, the largest overestimation and discrepancy also appear in the areas with higher latitude. Comparing to the variation with longitude showed in the figure below, the meridional distribution characteristics of error and bias are more significant and vary more consistently. Therefore, the variations of the statistical parameters with latitude are provided in Figure 6. Correlation coefficient, relative mean square error, normalized mean bias and normalized mean error between the simulations of MB, AG and KOK scheme and observed PM10.
Line 239 (and following): what does it means “mean correlation”?
It is the averaged correlation coefficient. Present line: 268.
Line 269: which two schemes?
They are MB and AG dust scheme. Revised into “The MB and AG scheme used the same….” Present line: 291.
Line 304: what does it means areal mean concentration?
It is revised into “regional averaged concentration”. Present line: 345.
We have tried our best to improve the manuscript and made changes in the manuscript. These changes will not influence the content and framework of the paper. And here we do not list the changes but marked in the revised manuscript. We thank the reviewer again for the constructive advice that have helped us to improve our manuscript.

Round 2
Reviewer 1 Report
The manuscript has not been significantly improved and is not scientific enough for publication in Sustainability.
Reviewer 2 Report
The authors provide the requested correction to the paper that can be considered for the publication.